# A FRET biosensor reveals spatiotemporal activation and functions of aurora kinase A in living cells

Giulia Bertolin[1,2], Florian Sizaire[1,2], Gaëtan Herbomel[1,2], David Reboutier[1,2,3], Claude Prigent[1,2,3] & Marc Tramier[1,2,4]

Overexpression of *AURKA* is a major hallmark of epithelial cancers. It encodes the multifunctional serine/threonine kinase aurora A, which is activated at metaphase and is required for cell cycle progression; assessing its activation in living cells is mandatory for next-generation drug design. We describe here a Förster's resonance energy transfer (FRET) biosensor detecting the conformational changes of aurora kinase A induced by its autophosphorylation on Thr288. The biosensor functionally replaces the endogenous kinase in cells and allows the activation of the kinase to be followed throughout the cell cycle. Inhibiting the catalytic activity of the kinase prevents the conformational changes of the biosensor. Using this approach, we discover that aurora kinase A activates during G1 to regulate the stability of microtubules in cooperation with TPX2 and CEP192. These results demonstrate that the aurora kinase A biosensor is a powerful tool to identify new regulatory pathways controlling aurora kinase A activation.

[1] CNRS, UMR 6290, Rennes 35043, France. [2] Université de Rennes 1, Institut de Génétique et Développement de Rennes, Rennes 35043, France. [3] Equipe labéllisée Ligue Contre Le Cancer 2014–2016, Rennes 35043, France. [4] Microscopy Rennes Imaging Centre, Biosit, Université de Rennes 1, Rennes 35043, France. Correspondence and requests for materials should be addressed to M.T. (email: marc.tramier@univ-rennes1.fr).

The cell cycle consists of a series of molecular events required to yield two daughter cells from one mother cell. To warrant the faithful duplication of the genetic material, the centrosomes operate as platforms for the nucleation of microtubules forming the bipolar spindle. Abnormalities in centrosome number, function or positioning cause the formation of defective spindles that induce the unfaithful repartition of sister chromatids at cell division, a cancer-causing condition known as aneuploidy[1]. The fidelity of centrosomal functions is controlled by the interplay of several molecular actors, including centrosome-residing and non-residing proteins that cooperate in promoting spindle assembly and stability. These proteins include mitotic kinases in charge of cell cycle progression[2] such as the serine/threonine kinase AURKA. This protein regulates the duplication and the maturation of the centrosomes, the correct timing for mitotic entry, the assembly of the mitotic spindle and cytokinesis[3]. These multiple functions of AURKA at mitosis are ensured by the physical interaction of the kinase with a wide variety of protein partners. The genetic amplification of AURKA and its overexpression at the mRNA and at the protein levels is frequently observed in epithelial cancers, and it is associated with an increased number of centrosomes, defective mitotic spindles and aneuploidy[3–5].

Considering the key role of AURKA in the maintenance of cell physiology, it is essential to understand its mode of activation and inhibition in vivo. Structural and biochemical studies in vitro have demonstrated that AURKA activates through auto-phosphorylation on Thr288 (refs 6–8). The activated kinase physically interacts with the microtubule-associated protein TPX2 (targeting protein for Xklp2), and it constitutes to date the most well-characterized mechanism to yield a fully active AURKA, capable of interacting with its various partners[7,9–13]. TPX2 is a microtubule-associated protein with no kinase activity per se; its physical interaction with AURKA buries the phosphate group on Thr288 in the ATP-binding pocket and prevents the access of phosphatases[7,10,14]. Although a great number of molecular partners of AURKA are known from protein–protein interaction studies in cells, the AURKA–TPX2 complex was the only one to be resolved with crystallography approaches, suggesting that AURKA interacts with its substrates in a dynamic manner[7,15].

In cells, AURKA associates with centrosomes at interphase and with the mitotic spindle during mitosis[3]. The expression levels and the catalytic activity of AURKA are dynamically regulated throughout the cell cycle and peak at metaphase to promote an efficient cell division[3]. When cells exit the mitotic programme, AURKA is rapidly ubiquitylated and degraded by the ubiquitin-proteasome system before a novel G1 phase[14,16,17]. At this stage, the kinase is present at very low and often undetectable levels; its abundance increases during the S phase, concomitantly with centrosome duplication and maturation[18]. However, it remains largely unexplored whether AURKA is enzymatically active during the G1 and S phases. This is mainly due to technical limitations as at present, the activation of AURKA and its catalytic activity can only be measured in vitro or in end-point assays in cells, and these approaches require the kinase to be heavily expressed and activated to measure its catalytic activity. Therefore, it was mandatory to develop new tools to follow the spatiotemporal activation of AURKA regardless of the expression levels of the kinase. Förster's resonance energy transfer (FRET)-based biosensors represent useful tools to address this issue, and they have been recently used to gain insight into the catalytic activity of mitotic kinases during cell cycle progression[19,20].

We here develop the first FRET-based biosensor of AURKA containing the full sequence of the kinase within a donor–acceptor fluorophore pair suitable for FRET. We demonstrate

that it measures the conformational changes of AURKA in vitro and in cellulo, and that these changes account for the activation of AURKA by autophosphorylation on Thr288. Furthermore, we provide a functional validation of the AURKA biosensor by showing that it replaces the endogenous kinase in living cells. The biosensor allows us to track the activation of AURKA in each phase of the cell cycle, and it responds to perturbations of the AURKA signalling pathway at mitosis as previously described for the endogenous protein[21,22]. Finally, the AURKA biosensor uncovers an activation of AURKA in G1. In this phase, AURKA is required to regulate the stability of the microtubule network in an interplay with TPX2 and the centrosomal protein of 192 kDa CEP192.

## Results

**In vitro validation of the AURKA FRET biosensor.** It is known that AURKA changes the conformation of its activation loop when it undergoes autophosphorylation on Thr288 (refs 7,15,23). We investigated whether this conformational change could be tracked in space and time by FRET microscopy. We fused a widely used donor–acceptor FRET pair to each terminus of AURKA: the enhanced green fluorescent protein (EGFP) donor fluorophore to the amino terminus and the mCherry acceptor fluorophore to the carboxy terminus (Fig. 1a)[24]. As FRET between the two fluorophores occurs only if the donor and the acceptor are in close proximity ($\leq 10$ nm), changes in FRET efficiency provide information on fluorophore orientation and help to infer the conformation of the protein[25,26]. We hypothesized that the modification of the ATP-binding pocket of AURKA brings the donor and the acceptor in proximity, allowing the measurement of FRET (Fig. 1a). We estimated the efficiency of FRET by using a fluorescence lifetime imaging microscopy (FLIM) approach, in which a donor molecule in proximity of an acceptor molecule shows a reduced fluorescence lifetime compared with the donor alone, due to the FRET effect[27]. We expressed and purified the GFP-AURKA-mCherry protein and the acceptor-devoid control GFP-AURKA from Escherichia coli, and we analysed their mean fluorescence lifetime by FLIM (Fig. 1b and Supplementary Fig. 1a); these measurements appeared homogeneous in space, with a s.d. of $\pm 50$ ps, which are intrinsic to the microscope system used for the measurement of FLIM. GFP-AURKA showed a mean lifetime comprised between 2,400 and 2,500 ps, which is compatible with the lifetime of the EGFP protein[24,26]. GFP-AURKA-mCherry had a mean lifetime spanning between 2,300 and 2,350 ps, ∼150 ps lower than the one of GFP-AURKA, indicative of FRET between EGFP and mCherry. Taking into account the dark species of mCherry and following previously published strategies to estimate FRET efficiency, we calculated that the ∼150 ps difference between GFP-AURKA and GFP-AURKA-mCherry corresponds to a FRET efficiency of ∼16% (ref. 24). To underline the relevance of FRET within the AURKA biosensor, it should be noted that a FRET-positive GFP-mCherry tandem probe in which the EGFP fluorophore is fused to mCherry, representing the condition of maximal FRET efficiency that can experimentally be obtained between the two fluorophores, shows a FRET efficiency of ∼22% (ref. 28). This comparison therefore highlights the power of the AURKA biosensor for the analysis of the conformational changes of the kinase.

As previously described, protein kinases issued from E.coli can be phosphorylated during the expression phase, and these modifications are often maintained during the purification procedure as it is the case for AURKA[29]. To gain insight on the correlation between FRET efficiency and phosphorylation, we dephosphorylated GFP-AURKA and GFP-AURKA-mCherry

with the generic Ser/Thr/Tyr phosphatase lambda (λPP). We treated the proteins for 1 h at 30 °C to ensure the optimal enzymatic activity of the phosphatase, and we measured the mean lifetime of the proteins all along the treatment (Fig. 1c). While the lifetime of GFP-AURKA remained constant at ∼2,500 ps, the

one of GFP-AURKA-mCherry started to increase after 35 min of treatment with λPP, and it reached the lifetime of GFP-AURKA after 50 min indicating a progressive loss of FRET between EGFP and mCherry during the dephosphorylation phase. We then incubated the dephosphorylated proteins with ATP at 30 °C for

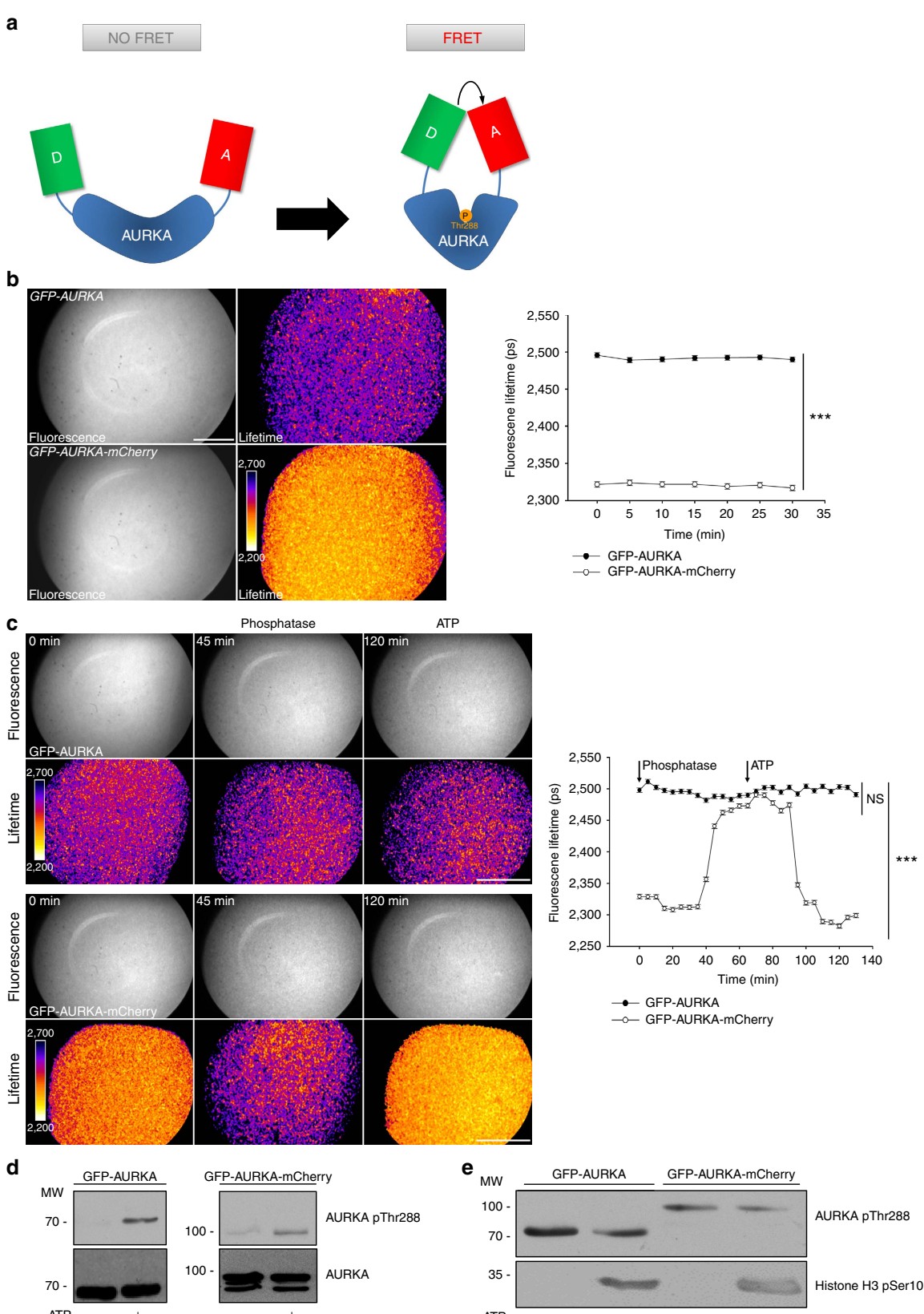

1 h to trigger their re-phosphorylation (Fig. 1c). While the lifetime of GFP-AURKA remained unaltered, the lifetime of GFP-AURKA-mCherry decreased from 2,500 to 2,300 ps after 25 min of treatment. We therefore concluded that the FRET observed within the AURKA biosensor is linked to the phosphorylation of the kinase. We also evaluated the effect of temperature on the efficiency of the dephosphorylation and re-phosphorylation reactions by treating GFP-AURKA and GFP-AURKA-mCherry with λPP and ATP at 37 °C, which is the standard temperature for ATP hydrolysis *in cellulo*. We measured the mean lifetime of the proteins throughout the treatments at 37 °C, and we observed that both reactions were completed faster than at 30 °C (Supplementary Fig. 1b). However, the dephosphorylation of GFP-AURKA-mCherry resulted less efficient at 37 °C, as the lifetime of the AURKA biosensor never reached the one of GFP-AURKA. This is due to the fact that λPP requires a temperature of 30 °C for an optimal catalytic activity, and it is partially inactivated when the temperature rises, as indicated by the manufacturer.

We then explored whether FRET within the AURKA biosensor is linked to the autophosphorylation of AURKA on Thr288 and to its kinase activity. We first performed standard *in vitro* kinase assays to estimate the autophosphorylation on Thr288 (pThr288) in samples dephosphorylated and re-phosphorylated as for FLIM analysis. We observed that a pThr288-specific band was absent when GFP-AURKA and GFP-AURKA-mCherry were treated with λPP, whereas this band appeared after the addition of ATP (Fig. 1d). We then used the phosphorylation of histone H3 on Ser10 (pSer10) as an indicator of the kinase activity of AURKA[30]. As expected, phosphorylation of histone H3 on Ser10 was detected only in the presence of ATP both for GFP-AURKA and GFP-AURKA-mCherry, showing that the two proteins are enzymatically active (Fig. 1e). To verify whether the temperature could alter the catalytic activity of the biosensor, we evaluated the abundance of the pSer10-specific band in samples incubated at 30 and at 37 °C. We observed that histone H3 was phosphorylated in a comparable manner at both temperatures (Supplementary Fig. 1c). Of note, 30 min were sufficient for an *in vitro* kinase reaction at 37 °C, as previously shown (Supplementary Fig. 1b). Together, our findings suggest that FRET between the EGFP-mCherry donor–acceptor pair of the AURKA biosensor reflects the autophosphorylation of the protein on Thr288 and demonstrate that the AURKA biosensor is a functional version of AURKA in *in vitro* assays.

**Biosensor activation is abolished by AURKA inhibitors**. To further evaluate the relationship between autophosphorylation on Thr288 and the kinase activity of AURKA, we specifically mutated the AURKA biosensor on Lys162 (Lys162Met) to obtain a kinase-dead variant of this protein previously shown to be incapable of performing substrate phosphorylation *in vitro*[31]. We then measured the mean lifetime of GFP-AURKA-mCherry and of the Lys162Met variant by FLIM. In contrast with GFP-AURKA-mCherry, the lifetime of the GFP-AURKA Lys162Met-mCherry variant was similar to the one of GFP-AURKA, showing lack of FRET between the EGFP-mCherry donor–acceptor pair throughout all the treatments (Fig. 2a). This was also confirmed by the absence of pThr288 on GFP-AURKA Lys162Met-mCherry in any condition (Fig. 2b). These data demonstrate that FRET within the AURKA biosensor shows a correlation with the autophosphorylation on Thr288 and to a catalytically active AURKA.

To gain further insight on how altering the catalytic activity of AURKA perturbs the conformational changes of the AURKA biosensor, we studied the effect of the pharmacological inhibitors MLN8237 and MLN8054 on FRET efficiency. Being small ATP competitors, these compounds specifically prevent autophosphorylation on Thr288 and abrogate the kinase activity of AURKA[32,33]. We therefore hypothesized that MLN8237 and MLN8054 could abolish autophosphorylation-related FRET. We pre-treated GFP-AURKA and GFP-AURKA-mCherry with λPP, and we imaged the proteins while triggering the re-phosphorylation of the kinase with ATP in conjunction with dimethylsulfoxide, MLN8237 or MLN8054 (Fig. 2c). We carried out FLIM reactions at 37 °C to ensure the optimal rate of ATP hydrolysis together with the most effective binding rate of the inhibitors according to *in cellulo* studies[32,33]. The mean lifetime of GFP-AURKA-mCherry treated with ATP and dimethylsulfoxide significantly decreased after 7 min, while no overall lifetime shift was observed in the presence of MLN8237 or MLN8054, or when using GFP-AURKA (Fig. 2c). Of note, a transient decrease in the lifetime of GFP-AURKA-mCherry was observed when the protein was treated with MLN8054, suggesting that the inhibitory effect of this compound is not immediate. We then analysed GFP-AURKA and GFP-AURKA-mCherry through standard kinase assays to validate the efficacy of MLN8237 and MLN8054 on the abundance of the pSer10 and pThr288 bands (Fig. 2d). Compared with controls, the abundance of both bands were lower when each of the two inhibitors was used, validating the activity of these compounds on the activation of AURKA and on the phosphorylation of a canonical substrate of the kinase. Together, our results indicate that the AURKA biosensor responds to the pharmacological inhibition of the kinase activity of AURKA, and provide further evidence that the autophosphorylation on Thr288 correlates with the kinase activity of AURKA *in vitro*.

**Figure 1 | The AURKA biosensor detects the autophosphorylation of AURKA on Thr288 *in vitro*.** (**a**) Model illustrating the mode of action of the AURKA biosensor. The complete sequence of AURKA is located between the donor (D, EGFP) and the acceptor (A, mCherry) fluorophores. When AURKA is autophosphorylated on Thr288, the kinase undergoes a conformational change bringing the donor and the acceptor in proximity and allowing FRET detection. Of note, the real three-dimensional orientations of the two fluorescent proteins are not known. (**b**) (Left panels) Representative fluorescence (GFP channel) and lifetime images from *in vitro* FLIM analysis of purified GFP-AURKA and GFP-AURKA-mCherry proteins. (Right panel) The graph illustrates a time-lapse analysis of the fluorescence lifetime of EGFP for both proteins. Images were acquired every 5 min. Data represent means ± s.e.m. of three independent experiments. (**c**) (Left panels) Representative fluorescence (GFP channel) and lifetime images taken at selected time points, and (right panel) corresponding quantification of the *in vitro* FLIM analysis of GFP-AURKA and GFP-AURKA-mCherry following λPP and ATP treatments. All treatments were performed at 30 °C and images were acquired every 5 min. The addition of λPP and ATP is indicated by an arrow on the graph. Data represent means ± s.e.m. of three independent experiments. The pseudocolour scale in **b,c** represents pixel-by-pixel lifetimes; conditions and/or time points are indicated in italics. Scale bar, 5 μm. ***$P < 0.001$ against each time point in the corresponding 'GFP-AURKA' condition; NS, not significant. Statistical tests: two-way ANOVA. (**d**) Representative *in vitro* kinase assay and corresponding western blot analysis showing the proportion of AURKA undergoing autophosphorylation on Thr288 (AURKA pThr288) in samples incubated at 30 °C with λPP for 1 h and then treated or not with ATP for 1 h at the same temperature. Loading control: total AURKA. (**e**) Representative *in vitro* kinase assay illustrating the presence of a Ser10-positive band on histone H3 after the incubation of GFP-AURKA and GFP-AURKA-mCherry with ATP for 30 min at 37 °C. Loading control: phosphoThr288 AURKA.

**The biosensor reflects intramolecular FRET**. To further explore the properties of the AURKA biosensor, we investigated whether its activation *in vitro* could be due to intermolecular FRET events occurring between adjacent GFP-AURKA-mCherry molecules. To this end, we compared the mean lifetime of

GFP-AURKA-mCherry with the one obtained by fusing the donor and acceptor fluorophores on distinct AURKA molecules, used in conjunction. These proteins were treated with λPP and ATP to follow the autophosphorylation of AURKA on Thr288 by FLIM (Fig. 3a). GFP-AURKA in the presence of

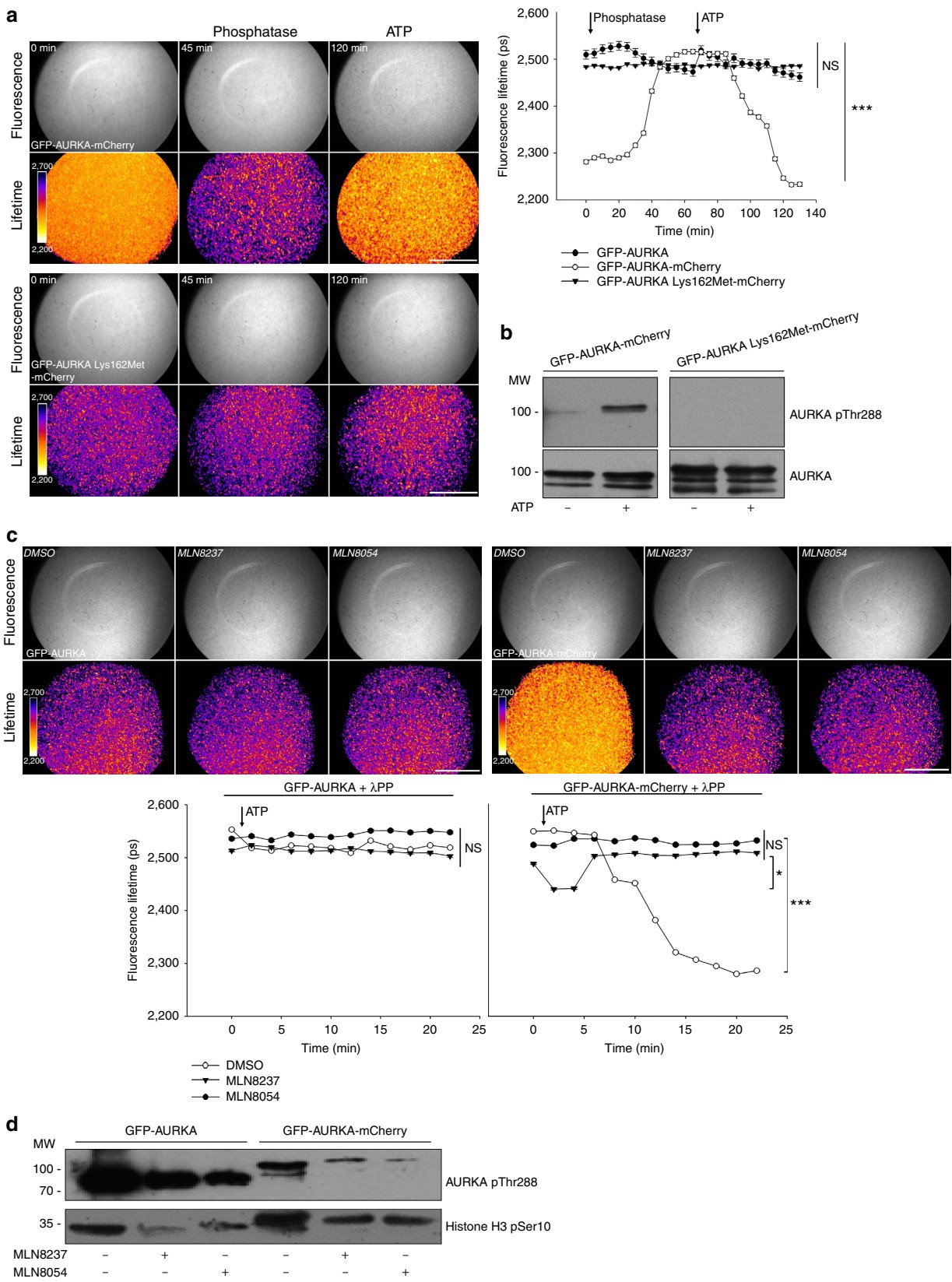

AURKA-mCherry showed a mean lifetime comparable to the one of GFP-AURKA alone throughout the treatments, thus not indicative of intermolecular FRET between two distinct donor–acceptor molecules.

As the AURKA biosensor is phosphorylated when purified from *E.coli*, we evaluated whether the incubation of GFP-AURKA-mCherry with increasing quantities of untagged AURKA could decrease FRET between EGFP and mCherry, indicating intermolecular FRET between neighbouring proteins (Fig. 3b). The lifetime of the biosensor remained unaltered for all conditions tested, strengthening the observation that the FRET due to the autophosphorylation of the AURKA biosensor on Thr288 follows an intramolecular mechanism.

**The biosensor functionally replaces endogenous AURKA.** It is known that the depletion of AURKA from cells induces spindle pole defects, an event representing a possible cause for aneuploidy[34–36]. We therefore asked whether the AURKA biosensor could functionally replace the endogenous protein by rescuing the centrosomal abnormalities detected after siRNA-mediated gene silencing of *AURKA*. Following a previously published strategy, silent mutations were inserted into the cDNA of GFP-AURKA and of GFP-AURKA-mCherry to become resistant to an *AURKA*-specific siRNA[37]. To overcome the problems issued from the massive overproduction of AURKA and to express the kinase at physiological levels, we stably transfected the vectors encoding GFP-AURKA or GFP-AURKA-mCherry under the control of the AURKA minimal transcriptional regulatory region in U2OS cells[37–41]. Knockdown of *AURKA* for 24 h in non-transfected U2OS cells resulted in nearly 60% of cells displaying multiple or monopolar spindle poles at metaphase (Fig. 4a and Supplementary Fig. 2a), whereas these features were only rarely detected in cells stably expressing GFP-AURKA-mCherry or GFP-AURKA. These data demonstrate that the biosensor fully rescues all previously described phenotypes[36,37].

**The biosensor activates according to its subcellular localization.** We then explored whether the AURKA biosensor could act as an efficient reporter of the activation of AURKA *in cellulo*. First, we explored the subcellular localization of the fluorescent proteins in the two cell lines at interphase and in mitotic cells and as expected, AURKA was found at centrosomes, in the cytosol and at the mitotic spindle (Fig. 4b and Supplementary Fig. 2b). Second, we evaluated the conformational changes of the biosensor in each of these subcellular compartments by comparing the lifetimes of GFP-AURKA and GFP-AURKA-mCherry. We measured a decrease of ∼150 ps in the lifetime of EGFP at centrosomes and at the mitotic spindle, which was comparable to the difference observed *in vitro* between GFP-AURKA and

GFP-AURKA-mCherry (Fig. 1b). In interphase cells FRET could be detected in the cytosol as well, although at lower levels (Fig. 4b). To further correlate FRET and the activation of AURKA, we analysed the inhibitory effect of MLN8237 on mitotic cells, in which the kinase activity of AURKA is greater compared with other cell cycle phases[42]. No significant FRET was measured on the mitotic spindle of cells treated with MLN8237 compared with controls (Fig. 4c). We also analysed the conformational changes of AURKA in the presence of the Lys162Met kinase-dead variant. We stably expressed this construct in U2OS cells following the same experimental strategy described above and similarly to the results obtained *in vitro*, no FRET was detected at centrosomes of interphase cells and on the mitotic spindle under these conditions. This provides further evidence that the activation of the biosensor in cells correlates with the enzymatic activity of AURKA (Supplementary Fig. 2c). In addition, FRET at centrosomes or at the mitotic spindle remained unaltered in control cells or following siRNA-mediated *AURKA* silencing, demonstrating that the activation of the biosensor is not affected by the knockdown of the endogenous kinase (Fig. 4d and Supplementary Fig. 2d).

**The biosensor differentially activates along the cell cycle.** To pursue the *in cellulo* characterization of the AURKA biosensor, we evaluated its conformational changes and its molecular dynamics in the different phases of the cell cycle. First, we synchronized GFP-AURKA and GFP-AURKA-mCherry cells in G1, S and at mitosis. The efficacy of the synchronization procedure was evaluated by flow cytometry (Supplementary Fig. 3a) and by the incorporation of the fluorescent ubiquitination-based cell cycle indicator (FUCCI) probe (Supplementary Fig. 3b). The measurement of the fluorescence intensity of GFP in GFP-AURKA and GFP-AURKA-mCherry cells after synchronization revealed that the expression of the biosensor is modulated during the cell cycle in both cell lines (Fig. 5a,b). FLIM analyses performed on cells synchronized at mitosis indicated that the biosensor is activated at the centrosome (Fig. 5c). We also detected a similar degree of activation at centrosomes of G1 cells, whereas no FRET was detected on the same structures during the S phase.

We then explored the relative motility of the biosensor at centrosomes of synchronized cells. Fluorescence recovery after photobleaching (FRAP) experiments showed that the recovery of the fluorescence of the biosensor was never complete in cells synchronized in G1 or at mitosis. This suggests the existence of a sub-population of GFP-AURKA-mCherry molecules not exchanging with the cytosol and therefore not renewed after the photobleaching procedure in these cell cycle phases (Fig. 5d). This was not observed on centrosomes of cells synchronized in the S phase, where the motility of the biosensor was already full after

---

**Figure 2 | Autophosphorylation on Thr288 is sufficient for a catalytically active AURKA biosensor in the presence of ATP.** (**a**) (Left panels) Representative fluorescence (GFP channel) and lifetime images taken at the indicated time points and (right panel) corresponding quantification of EGFP lifetime images taken every 5 min from GFP-AURKA, GFP-AURKA-mCherry or GFP-AURKA Lys162Met-mCherry samples treated λPP for 1 h at 30 °C, and then incubated with ATP for 1 h at 30 °C. Data represent means ± s.e.m. of three independent experiments. (**b**) *In vitro* kinase assay and western blot analysis showing the abundance of AURKA pThr288 in samples containing GFP-AURKA-mCherry or the Lys162Met variant following treatment with λPP for 1 h at 30 °C and then incubated or not with ATP for 1 h at the same temperature. (**c**) (Upper panels) Representative fluorescence (GFP channel) and lifetime images, and (lower panels) corresponding quantification of EGFP lifetime from GFP-AURKA or GFP-AURKA-mCherry samples treated with λPP for 1 h at 30 °C, and imaged following the addition of ATP together with dimethylsulfoxide (DMSO), MLN8237 or MLN8054 for 20 min at 37 °C. Images were acquired every 2 min. Data represent means ± s.e.m. of three independent experiments. (**d**) *In vitro* kinase assay and corresponding western blot illustrating the abundance of a Ser10-positive band on histone H3 and of autophosphorylated AURKA after the incubation of GFP-AURKA and GFP-AURKA-mCherry with ATP and DMSO, MLN8237 or MLN8054. Scale bar, 5 μm. Arrows: addition of λPP/ATP. Pseudocolour scale: pixel-by-pixel lifetime. *$P < 0.05$, ***$P < 0.001$ against each time point in the corresponding 'GFP-AURKA' condition in **a** or in the corresponding 'Phosphatase + DMSO' condition in **c**. NS, not significant. Statistical tests: two-way ANOVA.

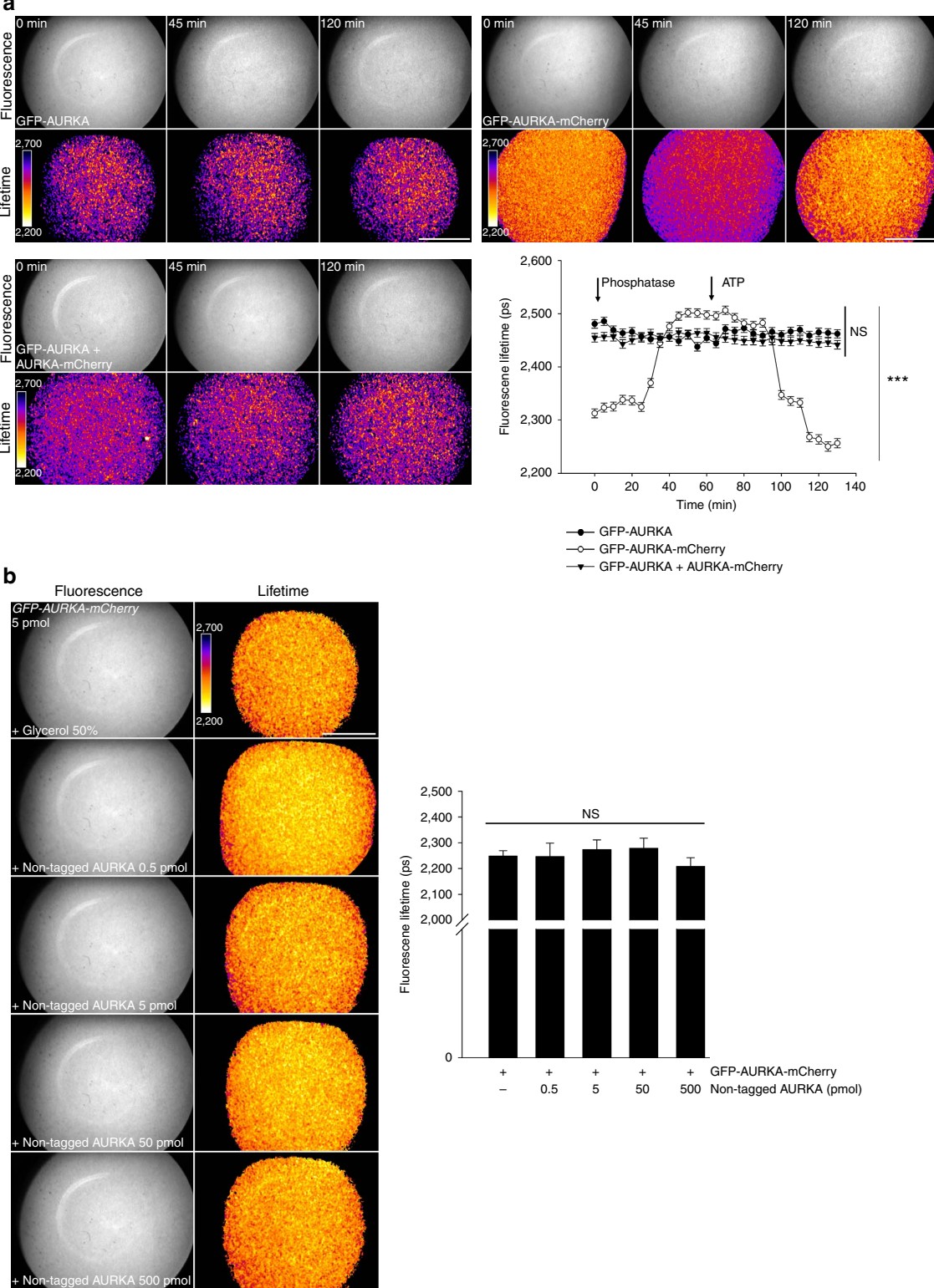

**Figure 3 | The AURKA biosensor displays intramolecular FRET *in vitro*.** (**a**) Representative fluorescence (GFP channel) and lifetime images taken at selected time points, and corresponding quantification of the *in vitro* FLIM analysis of the lifetime of EGFP from GFP-AURKA, GFP-AURKA-mCherry or GFP-AURKA and AURKA-mCherry samples incubated at 30 °C with λPP for 1h, and then treated with ATP for 1h. Images were acquired every 5 min. Data represent means ± s.e.m. of three independent experiments. ***$P < 0.001$ against each time point in the corresponding 'GFP-AURKA' condition. Statistical test: two-way ANOVA. Arrows: addition of λPP/ATP. (**b**) (Left panels) Representative fluorescence (GFP channel) and lifetime images of GFP-AURKA-mCherry incubated with the indicated quantities of untagged AURKA. (Right panel) Histograms show the corresponding quantification of the lifetime of EGFP obtained from the GFP-AURKA-mCherry protein treated as in the left panels. Data represent means ± s.e.m. of three independent experiments. Comparisons were made against the '+ glycerol 50%' condition. Pseudocolour scale: pixel-by-pixel lifetime. Scale bar, 5 µm. NS, not significant. Statistical test: one-way ANOVA.

60 s of recovery (Fig. 5e). Of note, no differences in FRAP were detected when using GFP-AURKA-mCherry cells or GFP-AURKA cells (data not shown).

Our data indicate that the AURKA biosensor has a differential activation and motility according to the cell cycle phase.

**Biosensor activation in G1 depends on TPX and CEP192.** We sought to verify whether the fraction of AURKA retrieved in G1 was enzymatically active, and not a non-degraded pool of the protein left after mitotic exit. To this end, we treated GFP-AURKA and GFP-AURKA-mCherry cells synchronized in G1

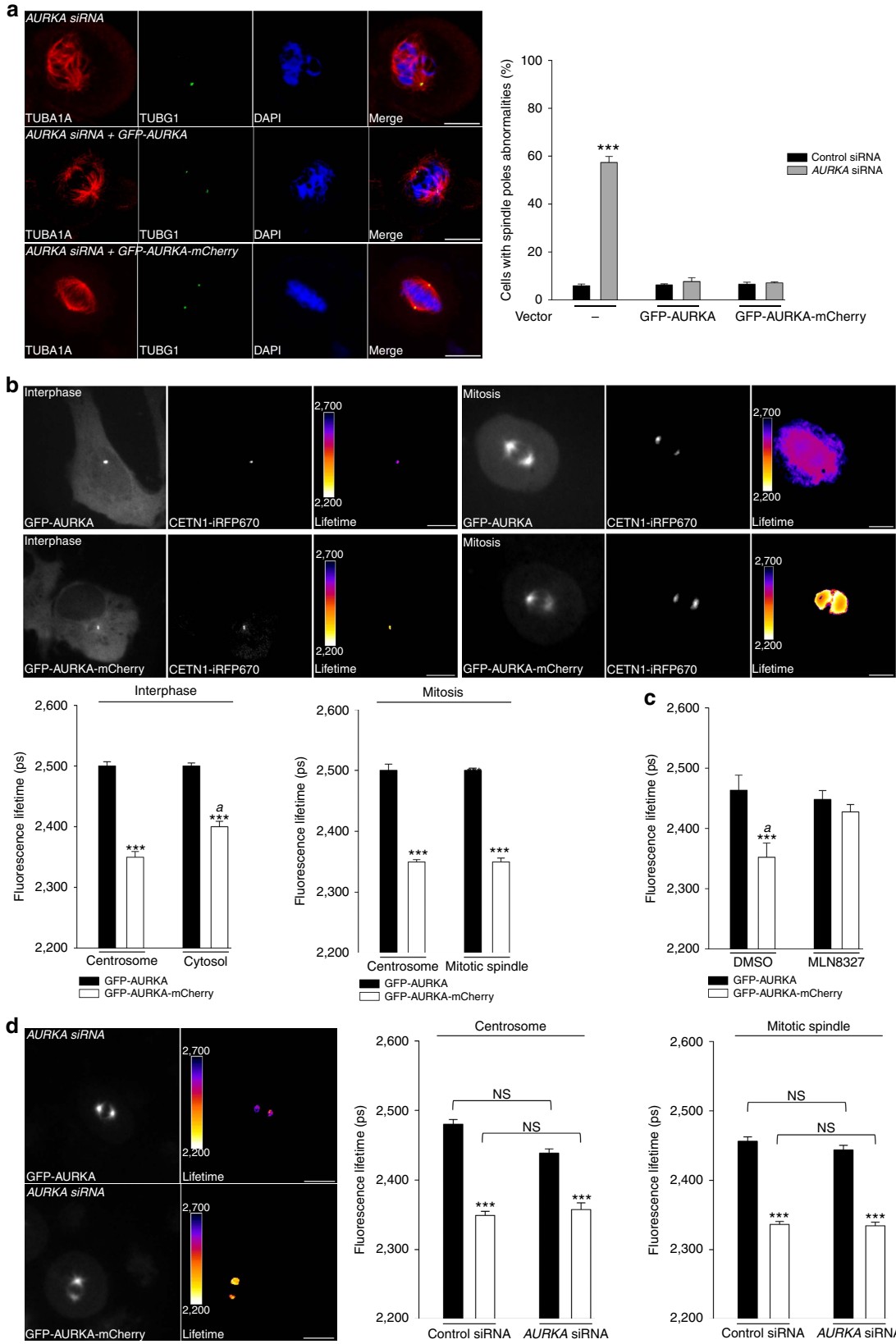

with the AURKA catalytic inhibitor MLN8237. Incubating cells with MLN8237 abolished FRET at centrosomes and indicated that the biosensor is enzymatically active in this cell cycle phase (Fig. 6a). This is also consistent with data obtained in mitotic cells under the same conditions (Fig. 4a).

To evaluate whether the biosensor at centrosomes could respond to perturbations in the AURKA signalling pathway, we silenced two well-characterized AURKA partners, TPX2 and CEP192. It is known that the downregulation of *TPX2* or of *CEP192* in mitotic cells induces abnormalities in centrosome maturation and in the formation of the mitotic spindle[21,22]. We retrieved these defects following siRNA-mediated gene silencing of both AURKA partners in GFP-AURKA and GFP-AURKA-mCherry cells synchronized at mitosis (Fig. 6b). FLIM measurements performed under these conditions showed that the lifetime of EGFP within the biosensor was higher at centrosomes of cells depleted for *TPX2* or *CEP192* compared with control cells (Fig. 6b). In addition, FRAP experiments were performed to explore the relative motility of the AURKA biosensor after the downregulation of *TPX2* or *CEP192* in mitotic cells. The fluorescence recovery of the biosensor was full after 30 s in cells depleted for *TPX2* or *CEP192*, whereas we observed the existence of a sub-population of GFP-AURKA-mCherry molecules not exchanging with the cytosol in control cells synchronized at mitosis (Supplementary Fig. 4a). These results suggest that the presence of TPX2 and CEP192 is required to anchor AURKA at the centrosome of mitotic cells (Supplementary Fig. 4a). Together, these data also reinforce the dependence of AURKA on TPX2 and CEP192 for its activation at mitosis as previously published[21,22], and validate the ability of the AURKA biosensor to detect the activation of the kinase in mitotic cells.

We subsequently explored whether the activation of AURKA at mitosis and in G1 follows similar molecular mechanisms. It is known that TPX2 is present in G1 cells and that its abundance is proportional to the abundance of AURKA throughout the cell cycle[21]. In addition, recent reports provided evidence that centrosome assembly features share common molecular mechanisms in G1 and in mitosis, and rely on CEP192 as a molecular scaffold[43,44]. We synchronized GFP-AURKA and GFP-AURKA-mCherry cells in G1 and we analysed the lifetime of EGFP at the centrosome following depletion of *TPX2* or *CEP192* by siRNA. The downregulation of both proteins abolished FRET between EGFP and mCherry at the centrosomes of GFP-AURKA-mCherry cells (Fig. 6c). Comparatively to what observed in FRAP experiments performed on mitotic cells, the silencing of *TPX2* and *CEP192* increased the relative motility of AURKA at the centrosome in G1

(Supplementary Fig. 4b). These data reinforce the conclusion that the AURKA biosensor is activated at centrosomes of G1 cells and it relies on the presence of TPX2 and CEP192 for its activation and its motility, as observed at mitosis.

**AURKA stabilizes microtubules in G1 together with TPX2 and CEP192.** As we provided evidence that TPX2 and CEP192 are needed for AURKA activation during the G1 phase, we addressed the functional role of this interplay. First, we noticed a marked amplification of CETN1 (Centrin-1)-positive spots in cells depleted for TPX2 and, to a lower extent, for CEP192 (Fig. 7a). A relation between an increased number of centrioles and altered microtubule dynamics has already been observed in mitotic cells[45,46], and further evidence was provided for the role of AURKA and TPX2 in the control of microtubule length during anaphase[47–49]. We therefore explored the role of TPX2 and of CEP192 in the stability of the microtubule network in GFP-AURKA-mCherry cells synchronized in G1. To this end, we evaluated the intensity of the TUBA1A staining in cells silenced for *TPX2* or *CEP192*. Under these conditions, the microtubule network appeared more fragmented and the TUBA1A staining was less intense than in control cells (Fig. 7b). To gain a dynamic view on how the formation of the microtubule network is affected in the absence of TPX2 or CEP192, we performed microtubule depolymerization and regrowth assays on HeLa GFP-TUBA1A cells depleted for *TPX2*, *CEP192* and *AURKA* and synchronized in G1 (Fig. 7c). By using time-lapse microscopy, we observed that microtubule regrowth remained incomplete in cells devoid of TPX2 and CEP192. Furthermore, the centrosomes of these cells were significantly bigger than control ones. Consistent with previous data obtained in mitotic cells[47–49], we found these alterations to be intrinsic to the AURKA signalling pathway in G1, as depletion of *AURKA* by siRNA gave similar results.

Altogether, our data provide compelling evidence for a role of an active pool of AURKA at centrosomes during the G1 phase, playing a role in the regulation of microtubule stability through the functional interaction with TPX2 and CEP192.

**Discussion**

We here provide evidence that the AURKA biosensor is a powerful tool to explore the activation of the kinase through the analysis of its conformational changes. We showed that these changes can be explored in real time and in specific spatial compartments by FRET microscopy, a sensitive technique widely used to detect transient protein–protein interactions and protein activity[25,26]. To gain insight into cell cycle progression, FRET biosensors were used to monitor selected kinases implicated in

**Figure 4 | The AURKA biosensor rescues the phenotype induced by *AURKA* deficiency and it is active at discrete subcellular locations.**
(**a**) (Left panels) Immunofluorescent micrographs of non-transfected, GFP-AURKA and GFP-AURKA-mCherry stable U2OS cells silenced for endogenous *AURKA* and synchronized in metaphase. The mitotic spindle and centrosomal defects in cells following *AURKA* depletion and the corresponding rescue by GFP-AURKA and by GFP-AURKA-mCherry were detected by labelling the mitotic spindle with TUBA1A and the centrosomes with TUBG1. DNA was stained with DAPI. (Right panel) Quantification of the proportion of cells with centrosomal defects in the three cell lines transfected with a control- or an *AURKA*-specific siRNA. $n = 100$ cells per condition scored in each of three independent experiments. (**b**) Representative fluorescence (GFP channel) and corresponding lifetime images of GFP-AURKA and GFP-AURKA-mCherry U2OS stable cell lines not synchronized (left panels) or synchronized in mitosis (right panels) and illustrating the presence of both proteins at the centrosome, in the cytosol and at mitotic spindles. Graphs: corresponding quantification of the lifetime of EGFP in the two cell lines and in the indicated subcellular compartments. Centrosomes were labelled with CETN1-iRFP670. $n = 30$–40 cells per condition from three independent experiments. (**c**) Quantification of the lifetime of EGFP GFP-AURKA and GFP-AURKA-mCherry cells synchronized at mitosis and treated with dimethylsulfoxide (DMSO) or with MLN8237. $n = 30$–40 cells per condition from three independent experiments. (**d**) (Left panels) Fluorescence (GFP channel) and lifetime images of GFP-AURKA and GFP-AURKA-mCherry cells transfected with an *AURKA*-specific siRNA and synchronized at mitosis. (Right panel) Quantification of EGFP lifetime in GFP-AURKA or in GFP-AURKA-mCherry cells transfected with a control- or an *AURKA*-specific siRNA and synchronized at mitosis; $n = 30$–40 cells per condition from three independent experiments. Data represent means ± s.e.m. Pseudocolour scale: pixel-by-pixel lifetime. Scale bar, 10 μm. \*\*\*$P < 0.001$ against the corresponding 'GFP-AURKA' condition in **a**–**d**; $^{a}P < 0.01$ against the corresponding 'GFP-AURKA-mCherry' condition at the centrosome in **b** and the corresponding MLN8327 condition in **c**. NS, not significant. Statistical tests: two-way ANOVA in **a**,**c** and **d**; Student's *t*-test in **b**.

the mitotic commitment such as CCNB1/CDK1, AURKB and PLK1 (refs 50–53).

Classic FRET biosensors contain a specific amino-acidic sequence targeted for modification, for example, the phosphorylation segment of a substrate protein, a ligand or a binding domain to favour the recognition and the folding of the target sequence, and the donor–acceptor FRET pair[19]. However, these tools present three major challenges: (i) the biosensor relies

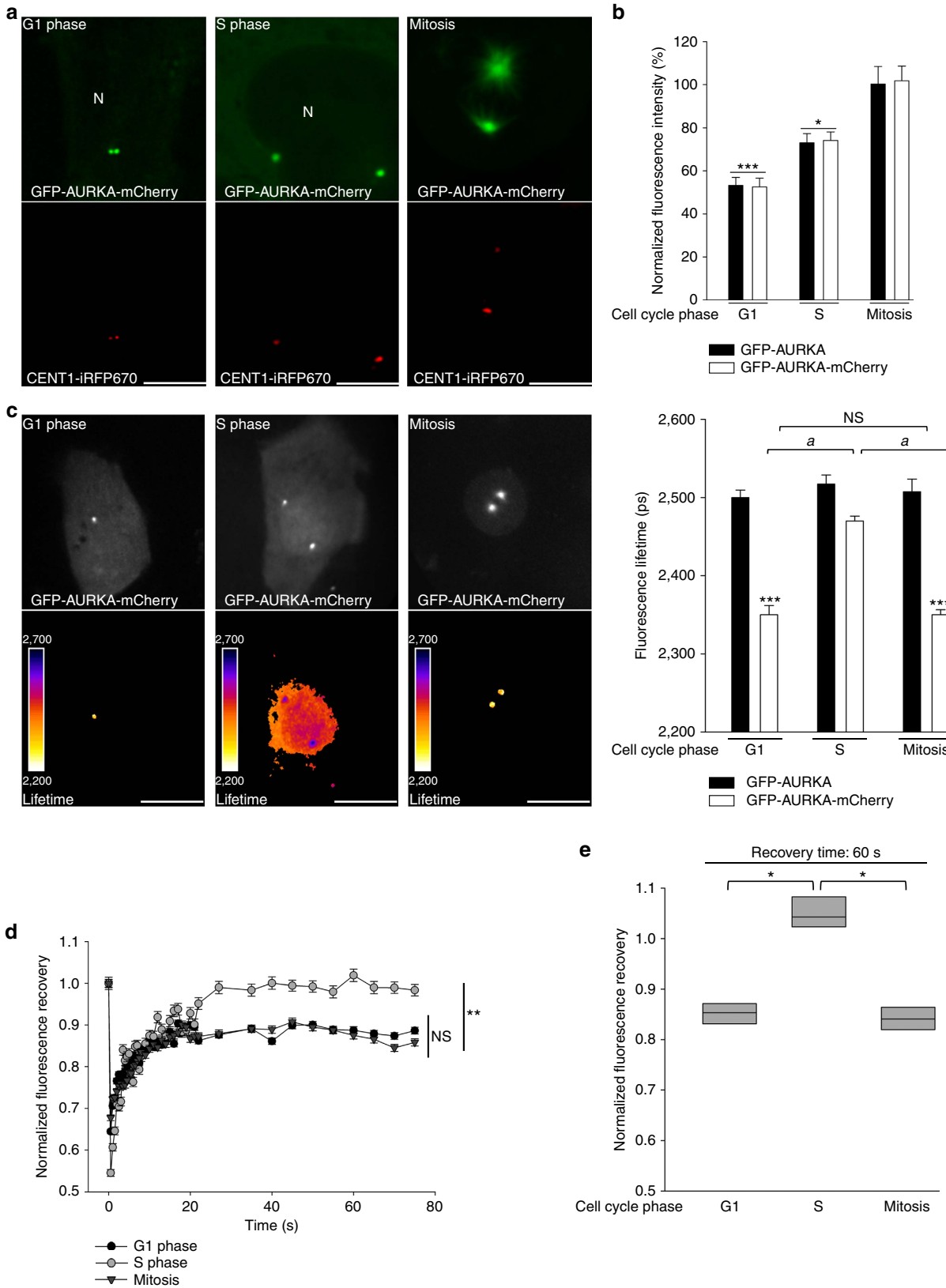

on the endogenous kinase for its phosphorylation and thus for FRET detection, which is observed only when the kinase is particularly abundant or heavily stimulated; (ii) the sequence flanking the phosphorylation residue(s) targeted by the kinase must be known; and (iii) these biosensors only explore the catalytic activity of the kinase towards a specific substrate at once, and not the conformational changes on the kinase itself. To solve this last issue, we developed a FRET biosensor of AURKA containing the full sequence of the kinase inserted between the donor–acceptor FRET pair. Similar approaches have previously been used to follow PRKGC and MELK, but the conformational changes observed by FRET were not directly assigned to kinase activation[54,55]. With the AURKA biosensor, we detected lifetime modifications in individual cells representing dynamic changes in the conformation of AURKA. In addition, we demonstrated that these changes correspond to the autophosphorylation of AURKA on Thr288, the hallmark of the molecular activation of the kinase. To date, the studies investigating the conformational changes of AURKA mainly rely on crystallography approaches. These reports indicate that only the ATP-binding pocket of the kinase modifies its conformation on autophosphorylation on Thr288, while the tertiary structure of the protein remains globally unaltered[7,9–13]. However, these studies were essentially performed with a trunked version of the kinase constituted only by its catalytic domain, and no or few information concerning the structure of the N and C termini is available. Importantly, the crystallography structure of AURKA in the absence of ATP remains unresolved, and this does not allow the comparison of the structures of the N and C termini between inactive and activated AURKA. Therefore, our data provide complementary information and demonstrate that both ends of the kinase are brought in proximity when AURKA is autophosphorylated on Thr288. As a large number of the crystal structures available to date show activated AURKA already in complex with ATP and TPX2, our knowledge on the physiological role of the autophosphorylation *per se* and by what molecular mechanisms this modification can take place is still fragmentary[7,10,23]. It is clear that this modification can occur independently of the presence of TPX2 (refs 6,10,15,56 and our own data). Remarkably, experiments performed in cultured cells showed that AURKA at the centrosome is not bound to TPX2 but it is phosphorylated on Thr288, whereas at the mitotic spindle the kinase binds to TPX2, but it shows poor or no autophosphorylation[57–59]. This underlines the importance of understanding the mechanisms controlling the autophosphorylation of AURKA and to this end, the AURKA biosensor is a sensitive tool to address this issue.

Furthermore, we provided strong evidence that cells regulate the expression of the AURKA biosensor as they would do for the endogenous protein[18,36,37,60]. Remarkably, this study constitutes the first case in which a biosensor takes over the functions of the

kinase in case of depletion of the endogenous protein. We also showed that the biosensor responds to the silencing of *TPX2* and *CEP192* at mitosis as previously published for endogenous AURKA[21,22], corroborating the functional integration of the biosensor in cells. Although TPX2 and CEP192 are nowadays the most extensively studied molecular partners of AURKA, it would be interesting to use the biosensor to characterize new or transient interactors of the kinase in their capacity to modify the activation of the biosensor at selected subcellular locations. In perspective, this tool could help to build a functional interactome of AURKA.

Given that AURKA levels are low during the G1 phase, the role of AURKA in maintaining microtubule stability could only be discovered through the activation of the biosensor at centrosomes in this cell cycle phase. The observation that the biosensor responds to the pharmacological inhibition of AURKA and to the depletion of *TPX2* and *CEP192* strongly supports a role for AURKA in this phase of the cell cycle. As the cell model used in this study is known not to produce primary cilia, we excluded the activation of AURKA in G1 to be linked to this function of the kinase in G0/G1 cells[61,62]. Interestingly, silencing of *TPX2* and, to a lesser extent, *CEP192* induces the apparition of multiple centrioles, a phenotype already described in mitotic cells when perturbing the AURKA–PLK4 signalling pathway and correlated with low microtubule dynamics[45,46]. Indeed, this study points at a new role of TPX2 and CEP192 within the AURKA signalling pathway in a previously unexplored cell cycle phase, and it provides another piece of evidence for the correlation between supernumerary centrioles and aberrant microtubule dynamics. Although it has been reported that cells can deal with the excessive number of centrosomes by clustering them and performing cell divisions with normal bipolar spindles to remain viable, it is known that wrong centrosomal clustering generates aneuploidy, and it ultimately results in cell death by apoptosis due to chromosome missegregation[63]. It cannot be excluded that AURKA plays a still unexplored role in the maintenance or in the efficiency of centrosome clustering through its functional interaction with TPX2, CEP192 and potentially other protein partners[64–66]. Targeting this interaction in G1 cells to perturb centrosome clustering in cancer-like conditions in which AURKA is overexpressed and it induces resistance to apoptosis could be a promising strategy to restore cell death[1]. However, further studies will be required to explore the partners of AURKA specific to this role of the kinase, as well as to deepen our knowledge on the molecular mechanisms of this regulation.

Finally, we propose that the changes in AURKA conformation could be used as an innovative read-out to screen for potentially new inhibitors of the kinase both *in vitro* and in living cells. The AURKA biosensor could also become a tool to explore the activation of the kinase in more complex *in vivo* models, paving the way to novel experimental approaches to address the

**Figure 5 | The AURKA biosensor activates during the G1 phase and at mitosis.** (**a**) Representative immunocytochemical images of GFP-AURKA-mCherry U2OS cells synchronized in the indicated cell cycle phases. The fluorescence of EGFP was used to visualize AURKA, centrosomes were labelled with CETN1-iRFP670. N, nucleus. (**b**) Quantification of the fluorescence intensity of EGFP in GFP-AURKA and GFP-AURKA-mCherry cells synchronized in G1 phase, S phase and at mitosis. The fluorescence intensity of both proteins at mitosis was arbitrarily set at 100% and it was used to normalize the fluorescence intensities of EGFP in the other conditions. $n = 40$–50 cells per condition from three independent experiments. Data represent means ± s.e.m. (**c**) (Left panels) Representative fluorescence (GFP channel) and lifetime images of GFP-AURKA-mCherry cells and (right panel) corresponding quantification of the lifetime of EGFP in GFP-AURKA and GFP-AURKA-mCherry cells synchronized as in **a**. $n = 30$–40 cells per condition from three independent experiments. Data represent means ± s.e.m. (**d**) Quantification of the fluorescence recovery of EGFP in GFP-AURKA-mCherry cells synchronized as indicated and subjected to FRAP analysis. $n = 30$ cells per condition from three independent experiments. Data represent means ± s.e.m. (**e**) Degree of fluorescence recovery of EGFP after 60 s from the photobleaching procedure. Scale bar, 10 μm. $*P < 0.05$, $**P < 0.01$ and $***P < 0.001$ against the corresponding 'mitosis' condition in **b,d**, the corresponding 'GFP-AURKA' condition in **c** and against the 'S' condition in **e**. $^{a}P < 0.001$ against the corresponding 'S' condition within the GFP-AURKA-mCherry paradigm in **c**. NS, not significant. Statistical tests: one-way ANOVA in **b,d** and **e**; two-way ANOVA in **c**.

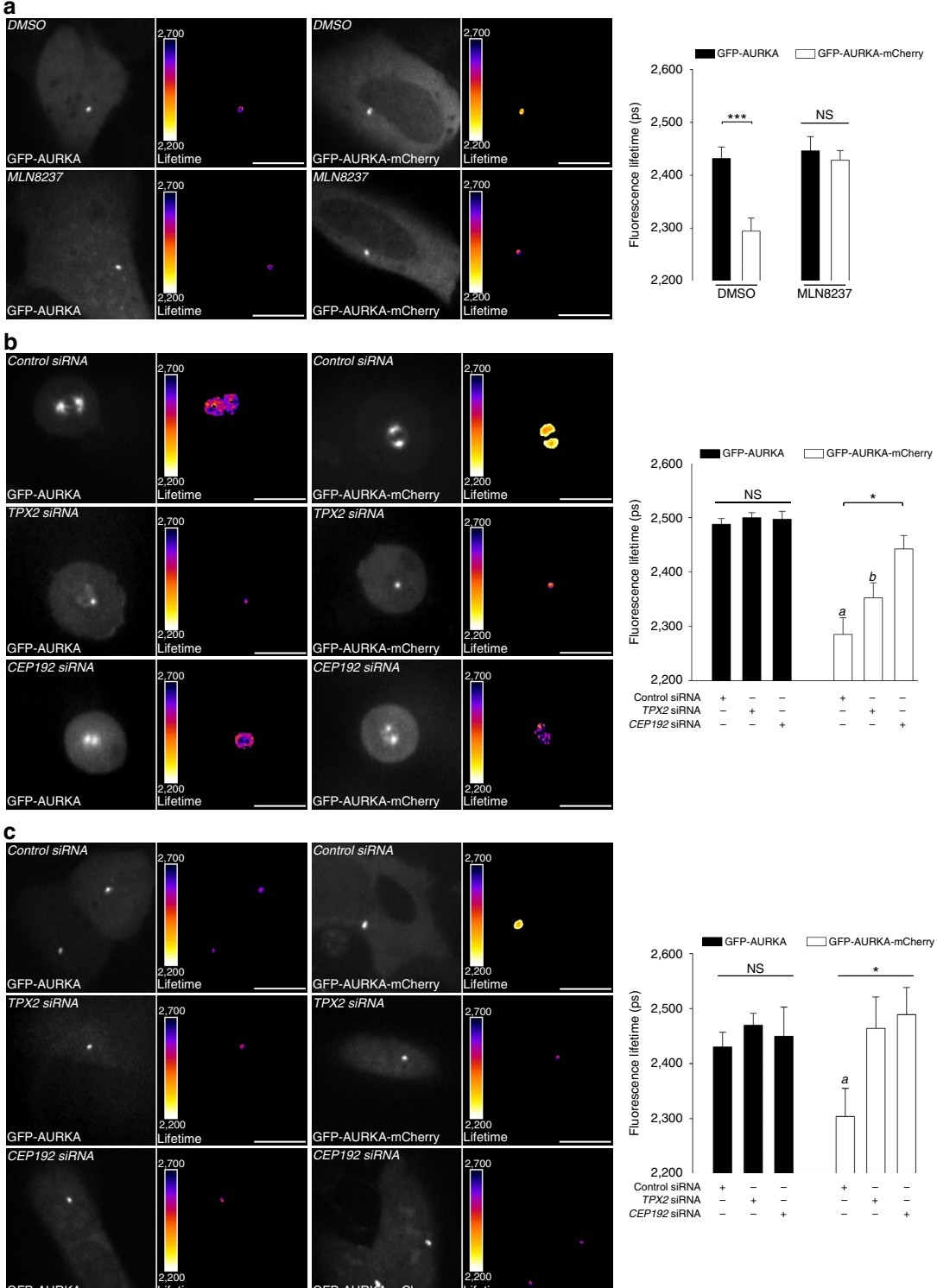

**Figure 6 | TPX2 and CEP192 are required for the activation of the AURKA biosensor during the G1 phase.** (**a**) (Left panels) Representative fluorescence (GFP channel) and lifetime images of GFP-AURKA and GFP-AURKA-mCherry cells synchronized in G1 and treated with dimethylsulfoxide (DMSO) or with MLN8237. (Right panel) Quantification of the lifetime of EGFP at the centrosomes of GFP-AURKA and GFP-AURKA-mCherry cells synchronized in G1 and treated as in the corresponding left panels. $n = 10–15$ cells per condition from one experiment representative of three. (**b**) (Left and middle panels) Representative fluorescence (GFP channel) and lifetime images of GFP-AURKA or of GFP-AURKA-mCherry cells transfected with a control, a *TPX2*- or a *CEP192*-specific siRNA and synchronized at mitosis. (Right panel) Corresponding lifetime quantification. $n = 10–15$ cells per condition from one experiment representative of three. (**c**) Images and quantification performed as in **b**, on cells synchronized in G1 and depleted for *TPX2* or *CEP192*. $n = 10–15$ cells per condition from one experiment representative of three. Data represent means ± s.e.m. Pseudocolour scale: pixel-by-pixel lifetime. Scale bar, 10 µm. ***$P < 0.001$ against the corresponding 'GFP-AURKA' condition in **a**, *$P < 0.05$ against the '*CEP192* siRNA' condition within the GFP-AURKA-mCherry paradigm and ${}^a P < 0.001$ and ${}^b P < 0.01$ against the corresponding 'GFP-AURKA' condition in **b**,**c**. NS, not significant. Statistical tests: two-way ANOVA in **a**; one-way ANOVA in **b**,**c**.

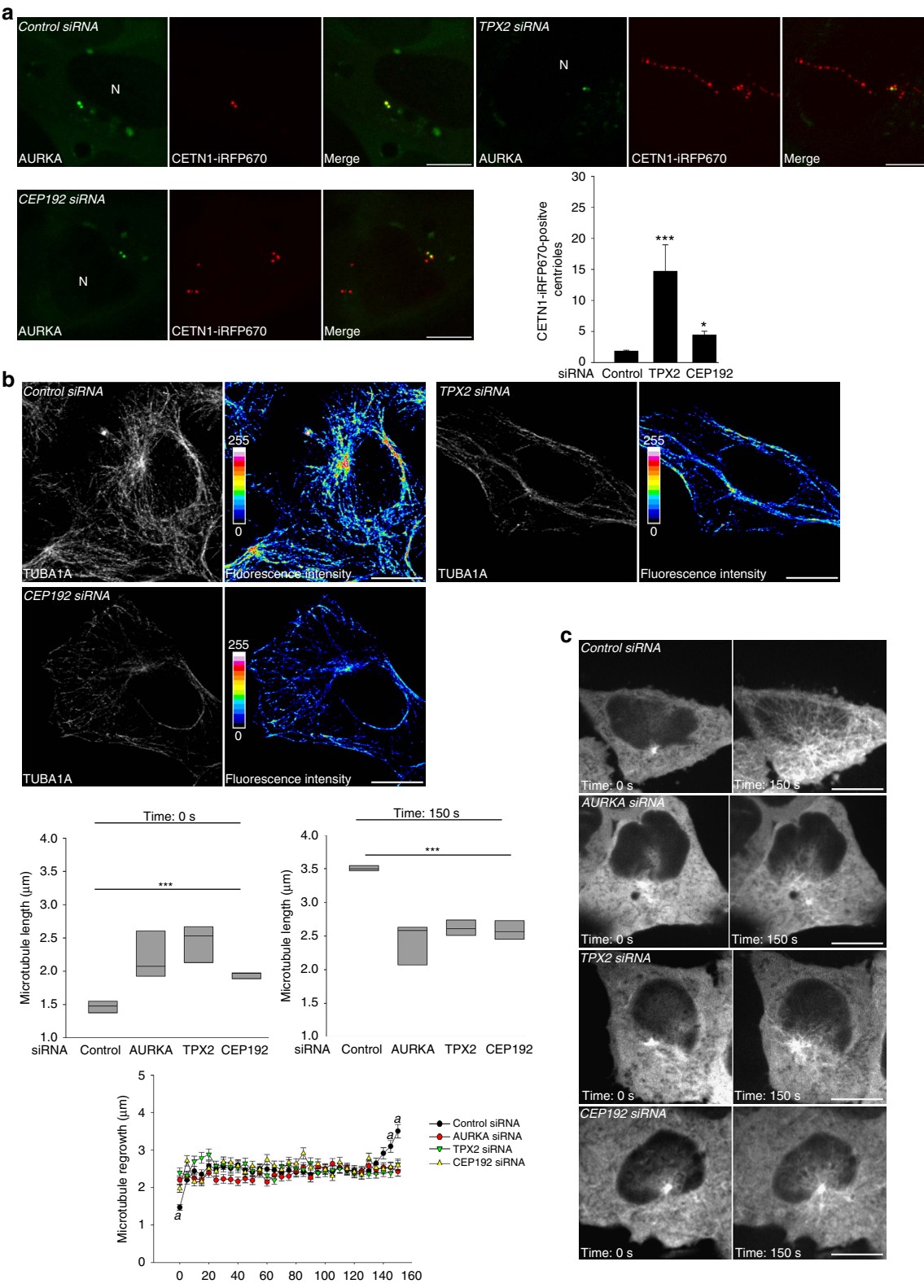

**Figure 7 | Microtubule dynamics of G1 cells is altered in the absence of TPX2 or CEP192. (a)** Representative immunofluorescence micrographs and corresponding quantification of centriole amplification in GFP-AURKA-mCherry cells depleted for TPX2 or CEP192, and synchronized in G1. Centrioles were labelled with CETN1-iRFP670. $n = 30$ cells per condition from three independent experiments. **(b)** Immunofluorescence images of the TUBA1A network in cells transfected and synchronized as in **a**. The pseudocolour scale represents the measure of the intensity of the TUBA1A staining. **(c)** Representative time-lapse images and corresponding boxplots of the regrowth of the microtubule network at 0 and 150 s after nocodazole-mediated depolymerization in HeLa GFP-TUBA1A cells transfected with the indicated siRNAs. Line graph: complete time-lapse analysis of the microtubule regrowth. $n = 30$ cells per condition from three independent experiments. N, nucleus. Scale bar, 10 µm in **a**,**b**, 5 µm in **c**. Data represent means ± s.e.m. *$P < 0.05$, *** $P < 0.001$ against the 'control siRNA' condition in **a**,**c**, $^{a}P < 0.01$ against the corresponding time-points in all other conditions. Statistical tests: one-way ANOVA.

functions of mitotic kinases both in physiological and in pathological conditions.

## Methods

**Expression vectors and molecular cloning procedures.** To express GFP-AURKA in bacteria, the coding sequence of EGFP was inserted between the BamHI/EcoRI sites of pET21a (Merck Millipore) and human *AURKA* between the SacI/SalI sites. For GFP-AURKA-mCherry, mCherry was cloned into the HindIII/XhoI sites in addition to the fragments used to build GFP-AURKA. For expression in mammalian cells, GFP-AURKA or GFP-AURKA-mCherry constructs were amplified by PCR from the corresponding pET21a vectors and subcloned into a pEGFP-C1 vector backbone (Clontech), where the CMV promoter was substituted by the *AURKA* minimal promoter sequence (CTTCCGG) into the AseI/BspEI restriction sites and the *AURKA* cDNA was mutated to become resistant to the *AURKA*-specific siRNA, following an already published strategy[37]. The AURKA Lys162Met variant was produced from pET21a GFP-AURKA-mCherry or from pEGFP-AURKA-mCherry by QuickChange site-directed mutagenesis (Stratagene) with the following primers: 5′-CAAGTTTA TTCTGGCTCTTATGGTGTTATTTAAAGCTCAGCT-3′ (sense) and 5′-AGCTG AGCTTTAAATAACACCATAAGAGCCAGAATAAACTTG-3′ (anti-sense). The plasmid encoding iRFP670 was obtained from Vladislav Verkhusha via Addgene and it has been previously described[67]. The cDNA of human CETN1 was a kind gift of J. Pécréaux (University of Rennes 1) and it was subcloned into the HindIII/BamHI sites of piRFP670-N1 to obtain CETN1-iRFP670. All cloning reactions were performed with the Gibson Assembly Master Mix (New England Biolabs) and verified on a 3130 XL sequencer (Applied Biosystems).

**Protein purification.** Recombinant AURKA proteins for in vitro FLIM analyses and kinase assays were obtained by transforming BL21(DE3) *E.coli* strains (New England Biolabs) with pET21a GFP-AURKA or GFP-AURKA-mCherry vectors. For protein production, bacteria were induced with 0.5 mM isopropyl β-D-1-thiogalactopyranoside for 16 h at 20 °C, centrifuged at 4 °C for 20 min at 4,000*g* and lysed in 1% Triton X-100, 500 µg ml$^{-1}$ lysozyme, 1 mM phenylmethylsulfonyl fluoride and protease inhibitors (Complete Cocktail, Roche). Bacterial lysates were then sonicated, centrifuged at 4 °C for 45 min at 13,000*g* and supernatants were incubated with Nickel-chelating resin (ProBond, Thermo Fisher Scientific) for 1 h at 4 °C under gentle rotation. Proteins bound to the resin were washed and eluted following the manufacturer's instructions for the purification of proteins under native conditions, and were then resuspended in 50% glycerol for storage at −20 °C. Histone H3 bound to glutathione-*S*-transferase was purified by the Protein Purification and Analysis Platform of the University of Rennes 1.

***In vitro* FLIM and biochemical kinase assays.** FLIM analyses on purified proteins and biochemical *in vitro* kinase tests were performed following a similar protocol. Briefly, 10 pmol of purified histone H3 and/or 5 pmol of purified GFP-AURKA or GFP-AURKA-mCherry were resuspended in 50 mM Tris-HCl (pH 7.5), 25 mM NaCl, 10 mM MgCl$_2$, 1 mM dithiothreitol and 0.01% Triton X-100. Where indicated, AURKA was dephosphorylated with 200 units of Lambda Protein Phosphatase (New England Biolabs) for 1 h, following the manufacturer's instructions. A unit of 100 µM ATP (Euromedex) was added and samples were incubated as follows: 1 h at 30 °C or 20 min at 37 °C for FLIM; and 20 min at 37 °C for biochemical kinase assays. MLN8237 and MLN8054 were purchased from Selleck Chemicals and used at final concentration of 50 and 30 nM. Total protein fractions were obtained from cells lysed in 50 mM Tris-HCl (pH 7.5), 150 mM NaCl, 1.5 mM MgCl$_2$, 1% Triton X-100, 0.5 mM dithiothreitol, phosphatase inhibitors (0.2 mM Na$_3$VO$_4$, 4 mg ml$^{-1}$ NaF and 5.4 mg ml$^{-1}$ β-glycerophosphate) and protease inhibitors.

**Western blot analyses.** Samples obtained from kinase tests or total protein fractions were boiled in Laemli sample buffer, resolved by SDS–PAGE, transferred onto a nitrocellulose membrane (GE Healthcare) and analysed by western blotting. The following primary antibodies were used: monoclonal mouse anti-AURKA clone 35C1, 1:20 (ref. 68) or rabbit anti-AURKA pThr288 1:2,000 (Thermo Fisher Scientific, MA5-14904); and anti-Histone H3 pSer10, 1:10,000 (Merck Millipore, 06-570) or rat anti-TUBA1A clone YL1/2, 1:5,000 (EMD Millipore, MAB1864). Secondary horseradish peroxidase-conjugated antibodies anti-mouse or -rabbit were purchased from Jackson ImmunoResearch Laboratories (315-035-045 and 111-035-144), while anti-rat were purchased from Bethyl Laboratories (A110-105P). Membranes were incubated with commercially available (Pierce) or home-made enhanced chemiluminescence substrate composed of 100 mM Tris (pH 8.6), 13 mg ml$^{-1}$ coumaric acid (Sigma), 44 mg ml$^{-1}$ luminol (Sigma) and 3% hydrogen peroxide (Sigma). Chemiluminescence signals were captured on a film (CP-BU new, Agfa Healthcare) with a CURIX 60 developer (Agfa Healthcare). Uncropped scans of western blots are shown in Supplementary Fig. 5.

**Cell culture and synchronization procedures.** Mycoplasma-free U2OS cells (HTB-96) were purchased from American Type Culture Collection and were grown in Dulbecco's modified Eagle's medium (DMEM, Sigma-Aldrich) supplemented with 10% fetal bovine serum (GE Healthcare), 1% L-glutamine (GE Healthcare) and 1% penicillin–streptomycin (GE Healthcare). GFP-AURKA and GFP-AURKA-mCherry cells were generated by transfecting U2OS cells with the corresponding mammalian expression vector and in the presence of the X-tremeGENE HP transfection reagent (Roche), following the manufacturer's indications. Stable clones were selected in DMEM supplemented with 10% fetal bovine serum, 1% L-glutamine, 1% penicillin–streptomycin and 500 µg ml$^{-1}$ geneticin (PAA). HeLa GFP-TUBA1A cells were a kind gift of C. Benaud (University of Rennes 1) and they were obtained by transfecting HeLa Kyoto cells with a plasmid encoding TUBA1A fused to GFP and in the presence of the jetPRIME transfection reagent (PolyPlus), according to the manufacturer's instructions. Stable clones were selected in DMEM supplemented with 10% fetal bovine serum, 1% L-glutamine, 1% penicillin–streptomycin and 500 µg ml$^{-1}$ geneticin. For live microscopy, cells were incubated in phenol red-free Leibovitz's L-15 medium (Thermo Fisher Scientific), supplemented with 20% fetal bovine serum, 1% L-glutamine and 1% penicillin–streptomycin. Validated siRNA against *TPX2* (SI02665082), *CEP192* (SI04164279) and AllStars negative control (SI03650318) siRNA were purchased from Qiagen; the siRNA against *AURKA* was synthesized as previously described[37] (sequence: 5′-AUGCCCUGUCUUACUG UCA-3′) and purchased from Eurogentec. siRNAs were transfected with Lipofectamine RNAiMAX (Thermo Fisher Scientific) for transfections with siRNA only, or with Lipofectamine 2000 (Thermo Fisher Scientific) for co-transfections with cDNA and siRNA, according to the manufacturer's instructions. GFP-AURKA and GFP-AURKA-mCherry U2OS cells were synchronized as follows: the synchronization at the G0/G1 transition was obtained by serum deprivation for 72 h. Cells were allowed to re-enter G1 phase for 2 h in normal growth medium before analysis. For synchronization in S phase, cells were synchronized with 5 µg ml$^{-1}$ aphidicolin (Sigma-Aldrich) for 18 h, followed by a 2 h washout in normal growth medium to re-enter S phase. Mitotic cells were obtained after synchronization at the G2/M transition with 100 ng ml$^{-1}$ nocodazole (Sigma-Aldrich) for 16 h. Cells were washed twice and incubated with prewarmed growth medium for 30 min to reach metaphase; FLIM, FRAP or flow cytometry analyses were performed after this stage. For microtubule regrowth assays, HeLa GFP TUBA1A cells were first synchronized in G1 as described above, then incubated on ice with 400 ng ml$^{-1}$ nocodazole for 20 min to induce TUBA1A depolymerization, quickly rinsed and imaged. MLN8237 was purchased from Selleck Chemicals and used at final concentration of 250 µM; cells were incubated with MLN8237 for 10 min before imaging. All live microscopy experiments were performed at 37 °C in Nunc Lab-Tek II Chamber slides (Thermo Fisher Scientific).

**Immunocytochemistry.** Cells were fixed 24 h after transfection with ice-cold methanol or with 4% paraformaldehyde (Sigma-Aldrich) where indicated, stained with standard immunocytochemical procedures and mounted in ProLong Gold Antifade reagent (Thermo Fisher Scientific). The antibodies used were as follows: primary monoclonal mouse anti-TUBG1 clone GTU88, 1:1,000 (Sigma Aldrich T6557) or rat anti-TUBA1A clone YL1/2, 1:1,000 (EMD Millipore, MAB1864); and secondary anti-mouse or anti-rat antibodies conjugated to Alexa 555 or Alexa 488, 1:5,000 (Thermo Fisher Scientific, A21434 and A11029, respectively). Hoechst 33342, Trihydrochloride, Trihydrate was purchased from Thermo Fisher Scientific and used at a concentration of 2 µg ml$^{-1}$. FUCCI Cell Cycle Sensor (BacMam 2.0) was purchased from Thermo Fisher Scientific and GFP-AURKA or GFP-AURKA mCherry cells were transduced according to the manufacturer's instructions.

**Widefield and confocal microscopy.** FUCCI transduction efficiency and cells displaying spindle poles abnormalities were scored with a DMRXA2 microscope (Leica), equipped with a × 63 oil immersion objective (numerical aperture (NA) 1.32) and driven by the MetaVue software (Molecular Devices). Multicolour images were taken with a Leica SP8 inverted confocal microscope (Leica) and a × 63 oil immersion objective (NA 1.4), driven by the LAS software. CETN1-iRFP670 spots were quantified with the Analyse Particles plugin of the ImageJ software.

**FLIM microscopy.** FLIM analyses were performed with a time-gated custom-built system attached to a Leica DMI6000 microscope (Leica) with a CSU-X1 spinning disk module (Yokogawa) and a × 63 oil immersion objective (NA 1.4), a picosecond pulsed supercontinuum white laser at 40 MHz frequency (Fianium), and a High-Rate Intensifier (LaVision) coupled to a CoolSNAP HQ2 charge-coupled device camera (Roper Scientific). Fluorescence excitation was selected at 480 ± 10 nm using a home-made wavelength selector. Fluorescence emission was selected with a band-pass filter (500–550 nm) (Semrock). To calculate fluorescence lifetime, five temporal gates with a step of 2 ns each allowed the sequential acquisition of five images covering a total delay time spanning from 0 to 10 ns (refs 24,69). The five images were used to calculate the pixel-by-pixel mean fluorescence lifetime according to the following equation: $\tau = \Sigma \Delta t_i \cdot I_i / \Sigma I_i$, where $\Delta t_i$ corresponds to the delay time after a laser pulse of the *i*th image acquired and *I* indicates the pixel-by-pixel fluorescence intensity in each image[24,69]. Lifetime measurements and calculations were performed with a custom-built programme running inside the MetaMorph software (Molecular Devices). Lifetime was calculated only when pixel-by-pixel fluorescence intensity in the first gate was above 3,000 grey levels.

**Fluorescence recovery after photobleaching.** FRAP analyses were performed on a Leica SP8 inverted confocal microscope with a ×63 oil immersion objective (NA 1.4) and using the Leica Acquisition Suite (LAS) software. A region of interest was photobleached with a 488 nm argon laser for 20 ms at maximum power. Fluorescence intensity images were taken before and after photobleaching with the same excitation wavelength; fluorescence recovery in the region of interest was followed until it reached a plateau (75 s) and a total of 40 frames were recorded every 500 ms (15 frames), 1 s (15 frames) and 5 s (10 frames). The ImageJ software (NIH) was used to measure fluorescence intensities. At least 30 images per condition were acquired and normalized as previously described[70].

**Microtubule regrowth assays.** Images of HeLa GFP-TUBA1A cells were taken with a Nikon Ti-E inverted confocal spinning disk microscope (CSU-X1 spinning disk scan head), a ×60 oil immersion objective (NA 1.4) and a sCMOS ORCA Flash 4.0 camera (Hamamatsu). Images of microtubules were acquired with a 491 nm laser and a 500–550 nm band-pass emission filter every 5 s, using the MetaMorph software. The quantification of the maximal length of microtubules at each time point and within each cell was performed applying and an automatic threshold over the microtubule area and using the Skeletonize plugin of the ImageJ software.

**Flow cytometry.** After phase-specific synchronization, GFP-AURKA and GFP-AURKA-mCherry U2OS cells were resuspended in 70% methanol and incubated at −20 °C for 20 min. Pellets were washed twice with PBS supplemented with 2 mM EDTA and centrifuged at 4 °C for 5 min at 2,500g after each washing. Cells were then resuspended in PBS-EDTA supplemented with 10 mg ml$^{-1}$ RNAse A (Euromedex) and 1 mg ml$^{-1}$ propidium iodide (Sigma Aldrich), incubated at room temperature for 20 min and the percentage of cells in each cell cycle phase was calculated with a FC500 flow cytometer (Beckman Coulter).

**Statistical analyses.** Two-way analysis of variance (ANOVA) and the Tukey method were used to compare the effects of the recombinant protein used and the duration of the treatment on fluorescence lifetime (Figs 1b,c, 2a and 3a, and Supplementary Fig. 1b). Two-way ANOVA and the Holm-Sidak method were used to compare the effects of the type of the inhibitor used and the duration of the treatment on fluorescence lifetime (Fig. 2c), the effect of AURKA-siRNA in cells stably expressing GFP-AURKA or GFP-AURKA-mCherry (Fig. 4a), the type fluorescent protein used in cells and the presence or absence of MLN8237 (Fig. 4c), the effect of AURKA-siRNA and the presence or absence of siRNA-resistant fluorescent AURKA (Fig. 4d), the effect of the fluorescent protein and the cell cycle phase on fluorescence lifetime (Fig. 5c), or the effect of the fluorescent protein and the presence or absence of MLN8237 (Fig. 6a). One-way ANOVA and the Holm-Sidak method were used to compare the effect of the indicated siRNAs on fluorescence lifetime (Fig. 6b,c) and on microtubule regrowth (Fig. 7c). One-way ANOVA on ranks and the Kruskal-Wallis methods were used to compare the fluorescence lifetime of GFP-AURKA-mCherry in the presence of increasing quantities of non-fluorescent AURKA (Fig. 3b), the effect of the Lys162Met mutation on fluorescence lifetime (Supplementary Fig. 2c), fluorescence intensity of GFP-AURKA-mCherry in cells synchronized in the indicated cell cycle phases (Fig. 5b), the rate of fluorescence recovery after photobleaching in synchronized cells (Fig. 5d,e) or after depletion of TPX2 or CEP192 by siRNA (Supplementary Fig. 4a,b), and the number of CETN1-iRFP670 spots after depletion of TPX2 or CEP192 by siRNA (Fig. 7a). The Student's t-test was used to analyse the effect of the fluorescent protein in each subcellular compartment analysed (Fig. 4b). Alpha for statistical tests used in this study was equal to 0.05.

**Data availability.** The data that support the findings of this study are available from the corresponding author on request.

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

## Acknowledgements

We thank S. Huet and T. Lebeaupin for helpful advice on FRAP experiments and their analysis, N. Mohamed for technical assistance, C. Benaud for providing HeLa GFP-TUBA1A cells together with X. Pinson, S. Le Bras, G. Rabut, E. Watrin and R. Le Borgne for helpful discussions. We are grateful to S. Prigent for valuable help in the analysis of microtubule regrowth and in macro designing, all the members of the Microscopy-Rennes Imaging Center (Biologie, Santé, Innovation Technologique, BIOSIT, Rennes, France), and Gersende Lacombe and Laurent Deleurme from the Flow cytometry and Cell Sorting platform (BIOSIT, Rennes, France) for assistance. This work was supported by Comité Nationale de la Recherche Scientifique, the Agence Nationale de la Recherche (ANR-11-BSV5-0023 KinBioFRET to C.P. and M.T.), by the Ligue Contre le Cancer Comité d'Ille et Vilaine, Comité du Maine et Loire et Comité de la Sarthe to M.T. and Equipe Labellisée 2014–2016 for D.R. and C.P., and by the Infrastructures en Biologie Santé et Agronomie (IBiSA) and Rennes Métropole for the development of the technology for rapid FLIM measurements. G.B. was supported by a fellowship from Fondation ARC pour la Recherche contre le Cancer.

## Author contributions

G.B. designed, performed and analysed the experiments and wrote the manuscript; F.S. performed and analysed the experiments; G.H. created the AURKA biosensor and performed essential preliminary experiments; D.R. and C.P. shared vectors, compounds and protocols and provided advice; M.T. supervised and coordinated the work and provided funding.

## Additional information

**Competing financial interests:** The authors declare no competing financial interests.

