## [Peer Review File · Nature Communications]

Reviewers' comments:

Reviewer #2, an expert in FRET probe design (Remarks to the Author):

The authors have developed a novel FRET biosensor for AURKA by fusing GFP and mCherry to the both ends of AURKA. First, the dephosphorylation and phosphorylation of the AURKA biosensor have been shown to correlate with the increase and decrease of fluorescent lifetime of EGFP, respectively. The decrease in fluorescent lifetime is caused by the phosphorylation of Thr288, i.e., activation of this protein kinase. The authors have also shown that the FRET is caused by intramolecular, but not intermolecular, association of the two fluorescent proteins. Then, the biosensor was expressed in U2OS cells with an AURKA minimal promoter. With this technique, physiological localization of the AURKA to the centriole has been achieved. The shortening of lifetime in the AURKA biosensor has been shown in the cells with the imaging data. Interestingly, they found that the lifetime of AURKA was increased during S phase, suggesting that AURKA activity was suppressed in S phase. FRAP analysis also showed that the immobile fraction of AURKA biosensor was almost gone in S phase. They also showed that TPX2 and CEP192 were required for AURKA biosensor activation.

A number of FRET biosensors have been reported for protein kinases. However, many of them including that of AURKA are the substrate-type biosensor to examine the protein kinase activity indirectly. Therefore, this AURKA biosensor is novel and important for the characterization of AURKA activity regulation. The authors are experts of FLIM and have convincingly shown that the new AURKA biosensor are very similar to the endogenous AURKA and probably reflect the activity of the authentic AURKA. This work has been well done in an organized way and adds new insights into the regulation of AURKA.

Suggested improvements are listed below:

1. Data presentation: Fig.1 to Fig. 3. For what purpose are the FLIM images presented? Unless there were any meanings in the distribution of lifetime in the images, they should be deleted.
2. Reproducibility: Fig. 1c: The authors claim that the lifetime of the probe increased 35 min after phosphatase addition and decreased 25 min after ATP addition at 37 centigrade; whereas the reaction is faster in 30 centigrade. It appears that both experiments were done just once. From the aspect of the enzymology, the data are rather weird. If the authors wish to discuss the time courses,

reproducibility should be examined. If this is just to show the effect phosphatase and in vitro kinase reaction, bar graphs will suffice to say the conclusion. Or, if the authors stick to show the sigmoidal curve of activation, mechanistic aspect should be discussed.

3. Presentation of data: "Fig. 4a: Scored in one experiment representative of three." The means of three experiments should be shown. The error bars here show the variation among the cell population. The variation among the experiments will be more important to show the robustness of the proposal.

4. Evaluation of data: Can the authors show clearly the fluctuation of FLIM data among different experiments. For example, in Figure 4, the lifetime of GFP-AURKA ranges from 2460 to 2500 ps. No doubt that the difference is statistically significant between GFP-AURKA and GFP-AURKCA-mCherry. But, the question is to what extent the authors could discuss in a quantitative manner.

5. Fig. 4b-d: Statistical significance must be examined among the GFP-AURKA-mCherry-expressing cells. It says the asterisks means the statistical difference against control siRNA condition. It does not look like so.

6. Fig. 5c: "FLIM analyses performed on cells synchronised at mitosis indicated that the biosensor is activated at the centrosome (Fig. 5c)." The authors must show FLIM images, because this is one of the most important data in this manuscript.

Reviewer #3, an expert in Aurora kinases and microtubules (Remarks to the Author):

This is a very nice paper. It describes a real need - an Aurora A sensor that will work in vivo and specifically in living cells. The experimentation has been very carefully controlled. The description of what exactly has been done is very clear - particularly important as this provides a useful reagent for the community. The sensor detects Aurora A activity in a manner expected from our knowledge of the enzyme and it has allowed the authors to identify new functions in G1 to regulate microtubule stability.

I strongly support publication of this paper.

Response to reviewers.

First review:

We are grateful for the reviewers' remarks and comments. The improvements suggested by reviewer 2 have carefully been taken into account and added to a new version of the manuscript.

Response to specific comments:

1. Significance of FLIM images in Fig. 1 to 3.

We agree with the reviewer that showing FLIM micrographs in these images is not mandatory to support our conclusions. However, we feel that it is important to show that the samples used under these conditions are homogeneously distributed. Although we agree that fluorescence images could be sufficient to this end, the presence of FLIM images corroborates the uniform distribution of lifetime in our samples. Importantly, FLIM images also show that the distribution of lifetime remains stable throughout time-lapse analyses (**Fig. 1, 2 and 3**), in the presence of the inactive Lys162Met mutant (**Fig. 2a**) or upon treatment with AURKA inhibitors (**Fig. 2b**). This strongly underlines that these procedures do not alter the homogeneity of the sample, but only intrinsic lifetime values.

FLIM images also show the pixel-by-pixel fluctuation of lifetime within each experimental condition and corresponding to the noise given by our FLIM system, regardless whether FRET occurs or not. This point is of particular interest for the reviewer (**point 4 of the present response**), and thus these images do reinforce the significance of FRET when it occurs *in vitro*. We therefore believe that this visual information is of interest to a broad readership, as for the one of *Nature Communications*. Of note, the homogeneity of FLIM measurements and its fluctuation have now been briefly described in the new version of the manuscript (**lines 115-117**). However, if the reviewer still feels that some of the data presented are not necessary to support our conclusions, we would be grateful to have precise indications on the dispensable figures.

2. Reproducibility of Fig. 1c.

In this point, the reviewer is concerned about two points: (a) the link between the speed of the enzymatic reactions and the temperature at which the reactions were performed, and (b) whether the experiment was performed once or in independent replicates.

a. Temperature and kinetics of phosphorylation/dephosphorylation.

We feel that the reviewer inverted the experimental temperatures that we used in Fig. 1c and Supplementary Fig. 1b. In **Fig. 1c**, both phosphorylation and dephosphorylation reactions were performed at 30°C. The kinetics are indeed slower at this temperature compared to the same experiments performed at 37°C, where the action of ATP is optimised and the reactions are faster (**Supplementary Fig. 1b**). We hope that the comparison of **Fig. 1c** and of **Supplementary Fig. 1b** will reconcile the reviewer with our enzymology data.

b. Data reproducibility.

Concerning this issue and as indicated in the corresponding figure legend, data in the graph of **Fig. 1c** represent means \pm s.e.m. of three independent experiments. To address the reviewer's concern, we improved the quality of the graph by reducing the size of the dots symbolising each time point. By doing so, the error bars representing the variation among the experiments become more visible.

3. Robustness of the rescue of the phenotypes induced by *AURKA* silencing.

The reviewer is concerned by the quantification of the phenotypes induced by the knockdown of *AURKA* and their corresponding rescue by the biosensor. As requested, we now provide in **Fig. 4a** a new version of the graph representing the variations among the three independent experiments performed, and not anymore among cell populations. The corresponding legend was updated as well (**lines 796-797**).

4. FLIM fluctuation among experiments

In this point, the reviewer aims at gaining a more detailed view on the fluctuation of FLIM values among the different experiments. The main difference is induced by the pixel-by-pixel fluctuation of lifetime as described in **point 1** of the present response. To give a more quantitative discussion of such fluctuation, in the graph below we show the lifetime of two GFP-*AURKA* cells (the control condition) issued from two replicates of the same experiment and performed on different days. Four images per cell were acquired successively (numbers 1 to 4 on the top of the graph). The calculation of the mean lifetime gives a value of 2508 psec for cell 1 and of 2460 psec for cell 2. We therefore estimated that these 50 psec of difference between the two cells represent the fluctuation of our custom-built system over time. To reinforce this conclusion, similar observations can be made for GFP-*AURKA*-mCherry, underlying that the fluctuation of the system is independent of the condition analysed. Finally, it must be noted that the difference in lifetime between GFP-*AURKA* and GFP-*AURKA*-mCherry is almost three times bigger, and this is of biological significance as demonstrated in the present manuscript. Again, the fluctuation of

FLIM measurements has been included in the manuscript (lines 115-117), as stated in point 1 of the present response.

5. Statistical significance of Fig. 4b-d.

We are grateful to the reviewer for his/her remark. Fig. 4b-d has now been updated, together with the corresponding figure legend (lines 810-812). Particularly for panel d, not significant comparisons (ns comparisons) were added to stress the observation that FRET remains unaltered after *AURKA* knockdown compared to control cells.

6. FLIM images at different cell cycle phases.

The images illustrating FRET at the centrosome for the *AURKA* biosensor (GFP-*AURKA*-mCherry) in the different cell cycle phases have now been integrated in the updated version of Fig. 5, panel c, together with the corresponding graph and legend (lines 821-824), as requested by the reviewer.

Second review:

The reviewer found our responses satisfactory, no further questions were added.